# Portable, low-cost samplers for distributed sampling of atmospheric gases

James F. Hurley[1], Alejandra Caceres[2], Deborah F. McGlynn[1], Mary E. Tovillo[1], Suzanne Pinar[3], Roger Schürch[3], Ksenia Onufrieva[3,4], Gabriel Isaacman-VanWertz[1]

[1]The Charles E. Via, Jr. Department of Civil and Environmental Engineering, Virginia Tech, Blacksburg, VA 24061, USA
[2]Bradley Department of Electrical and Computer Engineering, Virginia Tech, Blacksburg, VA 24061, USA
[3]Department of Entomology, Virginia Tech, Blacksburg, VA 24061, USA
[4]Center for Advanced Innovation in Agriculture, Virginia Tech, Blacksburg, VA 24061, USA

*Correspondence to*: Gabriel Isaacman-VanWertz (ivw@vt.edu)

**Abstract.** Volatile organic compounds (VOCs) contribute to air pollution both directly, as hazardous gases, and through their reactions with common atmospheric oxidants to produce ozone, particulate matter, and other hazardous air pollutants. There are enormous ranges of structures and reaction rates among VOCs, and consequently a need to accurately characterize the spatial and temporal distribution of individual identified compounds. Current VOC measurements are often made with complex, expensive instrumentation that provides high chemical detail, but is limited in its portability and requires high expense (e.g., mobile labs) for spatially resolved measurements. Alternatively, periodic collection of samples on cartridges is inexpensive but demands significant operator interaction that can limit possibilities for time-resolved measurements or distributed measurements across a spatial area. Thus, there is a need for simple, portable devices that can sample with limited operator presence to enable temporally and/or spatially resolved measurements. In this work, we describe new portable and programmable VOC samplers that enable simultaneous collection of samples across a spatially distributed network, validate their reproducibility, and demonstrate their utility. Validation experiments confirmed high precision between samplers as well as the ability of miniature ozone scrubbers to preserve reactive analytes collected on commercially available adsorbent gas sampling cartridges, supporting simultaneous field deployment across multiple locations. In indoor environments, 24-hour integrated samples demonstrate observable day-to-day variability, as well as variability across very short spatial scales (meters). The utility of the samplers was further demonstrated by locating outdoor point sources of analytes through the development of a new mapping approach that employs a group of the portable samplers and back projection techniques to assess a sampling area with higher resolution than stationary sampling. As with all gas sampling, the limits of detection depend on sampling times and the properties of sorbent and analyte. Limit of detection of the analytical system used in this work is on the order of nanograms, corresponding to mixing ratios of 1-10 pptv after one hour of sampling at the programmable flow rate of 50-250 sccm enabled by the developed system. The portable VOC samplers described and validated here provide a simple, low-cost sampling solution for spatially and/or temporally variable measurements of any organic gases that are collectable on currently available sampling media.

# 1 Introduction

Volatile organic compounds (VOCs) are emitted into the atmosphere from a wide range of sources, with vegetation as the main producer and anthropogenic sources accounting for roughly 10% of the emissions (Goldstein and Galbally, 2007; Guenther et al., 1995). VOCs and $NO_x$ (NO and $NO_2$) react in the presence of sunlight to produce photochemical smog (Haagen-Smit, 1950; Heald and Kroll, 2020), which is comprised of particulate matter (PM), ozone, and various other compounds detrimental to human health. High particulate levels strongly correlate with mortality and poor cardiovascular and respiratory health (Burnett et al., 2014; Dockery et al., 1993), while ozone is a powerful oxidant that adversely affects humans and does damage to crops, forests, rubber and other polymers (Felzer et al., 2007; Lippmann, 1989; Wark et al., 1998). Additionally, although VOCs lead to the production of criteria pollutants (ozone and PM), they are not monitored as regularly and methodically as the criteria pollutants. However, many VOCs, for instance small aromatics and halogenates, are known to be toxic to human health and in some cases are regulated as hazardous air pollutants (HAPs) (Tsai, 2019; US Environmental Protection Agency, 2023a, 2023b).

VOC emissions and concentrations are temporally and spatially diverse. Higher insolation and elevated daytime or seasonal temperatures promote the formation of oxidants (e.g., hydroxyl radical, ozone), the emission of biogenic VOCs, the volatilization of VOCs, and the enhancement of oxidant reaction rates, leading to variation in their sources and sinks on temporal timescales from minutes to seasons and concentration gradients across distances of only meters. These daily, short-term temporal variations occur in concert with shifts in emissions on the timescales of weeks (e.g., weekday-weekend effects in diesel/gasoline emissions) to months (e.g., changes in plant phenology) that yield a dynamic and highly variable mixture of VOCs. Spatial and temporal complexity is compounded by chemical complexity, with the number of biogenic VOC species having been estimated at over 30,000 (Fitzky et al., 2021; Trowbridge et al., 2013), though a small number of terpenoids dominate the mass of emitted carbon. A large fraction of anthropogenic emissions are composed of petroleum mixtures, which are comprised of hundreds or thousands of individual VOC species (Ilieş et al., 2021; Isaacman et al., 2012; Wang and Chen, 2017), and of hundreds of volatile chemical products (Steinemann, 2015). Each of these emitted compounds may react to form thousands to millions of products (Aumont et al., 2005), with an estimated $10^4$ to $10^5$ unique species present in the atmosphere at any one time and place (Goldstein and Galbally, 2007). To understand the chemistry or toxicity of the atmosphere, it is consequently necessary to measure a specific VOC or subset of VOCs within this dynamic mixture using methods that can capture their temporal and/or spatial variability.

There is a lack of available tools to make measurements or collect samples across an area to understand spatial distributions. Large, fixed, expensive instruments, such as mass spectrometers and gas chromatographs, record temporal variation with high chemical detail (Hamilton, 2010; Nozière et al., 2015; Yuan et al., 2017). These instruments may be put in mobile labs (for example, on an airplane) to get spatial information as well but require highly skilled operators and complex data analysis. In addition, the mobile labs may cover only larger spatial scales and are frequently limited to paved or otherwise constrained paths. In contrast, small handheld devices can capture both spatial and temporal variability, but these typically

lack any significant chemical resolution and in many cases are insufficiently sensitive for most atmospheric species of interest (Spinelle et al., 2017). Low-cost, spatially or temporally distributed measurements of VOCs frequently instead rely on collection of samples that are returned to a lab for offsite analysis to reduce the need for complex instrumentation in the field. This approach is low-cost and requires low operator effort but is also typically not autonomous or programmable;

sample collection requires the on-site presence of an operator, limiting sample collection to one point in space in time with an operator present at least at the sample start time. Distributed collection of coordinated samples would instead facilitate measurements of spatial gradients that would be useful in detecting leaks, pinpointing sources of hazardous or otherwise undesirable chemical compounds in occupational settings or understanding the spatial heterogeneity of atmospheric sources or sinks of tracer compounds.

Collection of air samples for offsite analysis usually follows one of two approaches. Whole air sampling collects a complete air sample into an inert bag or canister for analysis by a laboratory instrument, while sorbent sampling pulls air across a sampling cartridge containing an adsorbent polymer that traps compounds which then undergo thermal desorption (TD) or chemical extraction for later analysis. In either case, analysis commonly occurs by gas chromatography-mass spectrometry (GC-MS), which offers low limits of detection and high chemical selectivity. For sorbent sampling, different sorbents may

be used to target various VOCs of interest and large volumes of air can be sampled (Ciccioli et al., 1992; Yokouchi et al., 1990). Potential users of sorbent tubes should consult the literature and manufacturers to determine the most appropriate sorbent for their needs and target analytes, taking note of the compound types and size ranges adsorbed as well as potential artefacts (Borusiewicz and Zięba-Palus, 2007; Cao and Nicholas Hewitt, 1994; Hwan Lee et al., 2006; Klenø et al., 2002; Rothweiler et al., 1991). Advantages of sorbent tubes and TD over whole air samplers are the tubes' small size and

portability, lower cost, increased ability to transfer less volatile ($C_{10}$ and greater) and polar compounds, and greater ease of cleaning (Bianchi and Varney, 1993; Ciccioli et al., 1992; U.S. Environmental Protection Agency (EPA), 1999; Woolfenden, 1997). Drawbacks of the tubes are the limited ability to sample the most volatile compounds ($<C_4$), the presence of artefacts, the need to refrigerate unanalysed samples, and the requirement for a sampling pump (Betz and Supina, 1989; Bianchi and Varney, 1993; Ciccioli et al., 1992; Woolfenden, 1997). Furthermore, accurate measurements of absolute analyte

concentrations require knowledge of breakthrough volumes, i.e., the volume of sampled air that would completely elute the analyte through the sorbent and lead to underestimates (Harper, 1993). Due to their reusability, stability, and small size, sorbent tube sampling (sometimes referred to as "gas sampling cartridges") provides an attractive method for low-cost, portable sample collection. However, most current sorbent tube sampling technologies lack programmability, autonomy, portability, and/or computerized standardization (e.g., controlled flow rates) that would enable coordinated distributed

sample collection.

We present here a versatile, lightweight, battery- and solar-powered sampler capable of collecting atmospheric VOCs on sorbent tubes for GC-MS analysis, with simultaneous sampling across a network of samplers enabled by programmable sampling and weatherproof design. By enabling coordinated and self-powered sampling across multiple locations, these samplers fill a current gap in the need for affordable, lightweight devices capable of capturing spatial and temporal

variabilities (on scales as small as meters and as fast as ~hours) of analytes of interest in diverse settings. As will all analyses relying on sorbent cartridges or chromatography, sample analysis does require skilled personnel to operate the instrumentation and process the data, though the lower costs and mobility of the samplers should offer researchers much more latitude in sampling protocols. In this work, we describe the design of the sampler, demonstrate the reproducibility of sampling, and demonstrate the utility of a simultaneous network of multiple samplers, including a unique approach to

pinpointing an emission source across a spatially heterogeneous area.

## 2 Materials and Methods

### 2.1 Design and features of sampling boxes

Sampler boxes consist of a water- and air-tight polycarbonate case with a transparent lid, a visible LCD, and hidden

electronics (Fig. 1). Air is sampled through Teflon™ tubing connected to a metal adsorbent gas cartridge (3.5 in. × ¼ in. diameter) housed within the case. Sampled air is pulled by a variable-speed miniature diaphragm pump (Xavitech V200) and measured by a flowmeter (Honeywell Zephyr HAFBLF0200CAAX5) between the tube and pump; air is exhausted into the case, with a pressure relief valve to avoid overpressurization. A microcontroller with a real-time clock (PJRC Teensy® 3.5) receives flow measurements and provides a variable voltage to the pump to maintain a constant user-specified flow (50 to

300 ccm). Measured flow rate, case temperature, time until sample start, and elapsed sample time are stored on an on-board SD card and displayed to the user on a two-line LCD alongside a physical button for limited user interaction; the backlight on the LCD can be turned off during operation to conserve power. Communication between the pump, flowmeter, and microcontroller is managed by a custom circuit board with an onboard temperature sensor (Microchip MCP9700), with firmware written in the Arduino IDE. Easy physical access to communicate with the microcontroller is provided by a

microUSB port next to the LCD to easily update the firmware. Changes to flow rate and synchronization between samplers can be achieved through minor firmware updates, with readily available modes for: immediate start, start after a user-specified elapsed time, start at a specific time of day, or start at a specific date and time.

Power is supplied by a rechargeable 5V 12,800 mAh battery (i.e., USB power pack, Voltaic Systems V50) that powers the sampler for 49 hours at a higher sample flow (250 ccm) or for 65 hours at low sample flow (50 ccm). The battery is designed

to be charged by solar panels while in use to extend sampling (potentially indefinitely). The case and external components are all at least IP66-rated, ensuring the samplers are rugged and weatherproof; successful deployment in this work occurred under both wet and freezing cold conditions. Because sampled air is exhausted into the box and out through the pressure release valve (Fig. 1), residence time of air in the box is only a few minutes and the internal box temperature is maintained near ambient temperatures. The sampler dimensions are approximately 14 cm x 22 cm x 10 cm and weigh 1 kg. Most of the

size and weight of the sampler is due to the ruggedized box and high-capacity battery; the electronic components are all

mounted on a panel within the case that can be removed to create a serviceable 250 g version that has been tested for short-term deployment on small mobile platforms such as drones. In addition, many of the components used in this work are suitable for the development of a multi-channel sampler prototype, which was built and tested to allow sequential sampling but is outside the scope of this manuscript, which focuses on distributed and portable sampling.

## 2.2 Analysis of samples

Gas sampling cartridges consist of commercially available stainless-steel tubes packed with one or more adsorbing polymers (Tenax® TA, Tenax® GR, Carbotrap® 202, obtained from Sigma-Aldrich). These sorbents target broad ranges of mid- and lower-volatility gases and can be used to validate the samplers across a wide range of gases, including the more moderate-volatility gases that are more likely to suffer sampling losses on surfaces. Cartridges were thermally desorbed for analysis (TD-100-xr, Markes International) by gas chromatography and mass spectrometry (GC/MS; Focus GC/DSQ II MS, Thermo Scientific). Tubes were thermally desorbed at 250°C for 8 minutes, with a desorption flow of 50 mL/min and a flow path temperature of 190°C. Due to nearly constant use of the sorbent cartridges, it was determined that analysis of the tubes was equivalent to a cleaning step; however, tubes that had not been analysed for more than a few days were conditioned before use. Analytes were refocused on a cold trap at -30°C before subsequent re-desorption at a temperature of 320°C for 4 minutes to the GC. Separation was achieved using a nonpolar column (DB-5 phase, 30m x 0.25mm x 0.25µm) followed by detection by a unit-mass resolution quadrupole mass spectrometer (scan rate = 5 Hz, mass range = 33 – 350 amu). The GC temperature program was held at 40°C for 4 minutes, ramped at 15°C/min to 300°C, then held at 300°C for 5 minutes for a total run time of 26.3 minutes. For analysis of liquid standard mixtures, dichloromethane (DCM) or n-hexane served as solvents and a 2.5-minute solvent delay was used to avoid detector saturation. Mass spectra of the eluting peaks were identified by comparison with the NIST mass spectral library (Wallace, 2019) and further corroborated by their retention times against an alkane standard (Supelco Product # 04070).

Reproducibility and variability of sampled analytes can be measured based on integrated chromatographic peak area assuming linear response of the MS, so calibration of relative signal to yield quantitative ambient concentrations is not a focus of this work. However, calibrated ambient concentrations are reported for the spatial variability experiment (see Section 2.4.2) to demonstrate that introduced emissions sources would be generating atmospherically relevant concentrations for measurement. In all analyses, a blank sample was analysed consisting of a cartridge on which no sample was collected, and the analyte peak areas from the sample tubes were corrected by subtracting the peak areas (if any) observed in the blank. Compounds known to be commonly occurring artefacts of the used sorbent, such as straight-chain aldehydes and phenyl-substituted carbonyls (Hwan Lee et al., 2006; Klenø et al., 2002), were not included in any analyses.

## 2.3 Sampler validation

Samplers were examined to quantify the reproducibility of samples (i.e., the extent to which samples collected on different samplers are comparable) and evaluate the efficacy of pre-sampling treatment for ozone removal. The latter has been

previously shown to be necessary for sampling of reactive organic gases, so here we examine the feasibility of doing so using components similar in scale and cost as the rest of the sampler.

### 2.3.1 Sample reproducibility

Reproducibility between samplers was quantified as the relative standard deviation (% RSD = $100 \times$ SD/mean) of ambient analyte signal sampled by a set of co-located samplers. Twelve samples were simultaneously collected on Tenax® GR cartridges in the same indoor environment, separated into two clusters (n=5 and n=7) located approximately 2 meters apart. Samples were collected at 250 ccm for 7.5 hours in an active research lab with all windows closed. Reproducibility between samplers was quantified as % RSD within each set.

### 2.3.2 Ozone removal

Experiments were conducted to assess the effectiveness of ozone scrubbers, which preserve analytes susceptible to ozonolysis (e.g., alkenes). Ozone scrubbers were affixed to the inlet port of each box, comprised of filters saturated with sodium thiosulphate (Pollmann et al., 2005). Approximately 5.1 g of sodium thiosulphate ($Na_2S_2O_3$) was added to nanopure water for a 9.3 % (w/w) solution. With a Luer Lock syringe (Hamilton), 7 mL of the solution was pushed through a 25 mm diameter glass fibre filter with 1µm sized pores in a polypropylene housing (Acrodisc® Syringe Filter). The wetted filters were dried by placing them in an oven a 50°C and flowing 100 ccm nitrogen through them for 4 hours.

Two sets of samples were collected to examine conditions representative of ambient outdoor and indoor atmospheres. In the outdoors experiment, 10 samples were collected in front of open laboratory windows in Blacksburg, VA, with half the samplers using an ozone scrubber. Samples were collected for 8.7 hours at a sampling rate of 300 ccm. Indoor samples were collected following the same procedure, with all samplers in an interior (i.e., no windows) climate-controlled room. Indoor samples were collected for 13.5 hours at a sampling rate of 300 ccm. All samples were collected on Carbotrap® 202 cartridges.

Two families of compounds were analysed: BTEX (benzene, toluene, ethylbenzene, and three xylene isomers) and monoterpenes (α-pinene, camphene, β-pinene, 3-carene, and limonene, with the unsaturated ketone sulcatone also included in the outdoor dataset) respectively represent low and high reactivity with ozone. After a Jarque-Bera Test to ensure normality, analyte peak areas between scrubbed and unscrubbed samples were subject to a t-test (95% Confidence, two-tailed, homoscedastic) to detect statistical significance. Fractional analyte losses in unscrubbed samples were compared to ozone reaction rate constants ($k_{O3}$) on the NIST Chemical Kinetics Database, choosing more recently published experimental values (Manion et al., 2015).

## 2.4 Demonstration of sample collection

### 2.4.1 Long-term temporal variability

Samplers were deployed to determine the ability to measure day-to-day variability of commonly occurring VOCs, tested in an indoor setting. The goal of these experiments is to evaluate the reproducibility of samples (i.e., sampler-to-sampler variability) in the context of real-world temporal variability (i.e., time-to-time variability). A reasonably small % RSD between samplers compared to real-world variability would be necessary to discern statistically significant temporal variability.

Samples were collected on a tabletop in an infrequently used office over a three-week period in March-April 2022, thrice a week for 9 sets of measurements. For each set, three samples and a field blank were collected on Tenax® GR cartridges at a flow rate of 250 ccm for roughly a full day (1440 minutes). Day-to-day variability was quantified for various analytes of known environmental significance (e.g., BTEX, monoterpenes) and not known to be artefacts from sorbent decomposition. Analyte peak areas were corrected by subtracting the blank analyte peak area (if any) and normalized to sampling time (which was not identical for each set).

### 2.4.2 Spatial variability

To validate samplers' ability to record spatial heterogeneity in ambient concentrations at multiple spatial scales, two deployments were conducted. To investigate fine scale spatial difference, the samples collected for reproducibility tests (Section 2.3.1) were also examined for statistically significant differences (two-tailed t-test) between the two sets of samplers separated by ~2 meters in an indoor environment. Larger-scale heterogeneity was investigated by collecting samples within a 400 $m^2$ area to which known emission sources were introduced. We also develop here a novel approach to use back projection of sampled transects using mobile sampling devices to locate emissions sources within a gridded region with higher resolution than can be achieved by stationary devices.

To examine the utility of distributed sampling to locate emissions sources, the target area was split into a grid of $n$ x $n$ cells. Fig. 5 shows a generalized n x n sampling grid with the cardinal directions forming the two sampling angles of 0° and 90°. A three-angle grid would have sampling angles of 0°, 60°, and 120°; a four-angle grid would have angles of 0°, 45°, 90°, 135°, and so on. See Supplemental Section S4 for further discussion of the number of angles and optimization of the transect mapping approach.

Each cell can be probed by placing a sampler within the cell, requiring the collection of $n^2$ samples. To reduce the number of needed samples, however, spatially and temporally integrated samples can be collected in transects across the sample area. By collecting $n$ transect samples in two orthogonal directions, a pair of integrated concentrations is measured at the junction of each transect, providing $n^2$ unique datapoints describing the grid while collecting only $2n$ samples. Back projection enables these spatially integrated data to be allocated within the transected area. Here we apply this concept to identify the highest concentrations and thus the most likely point source location, an approach we describe here as "transect mapping" to

examine its utility and limitations. Larger numbers of transect angles can improve accuracy and reduce artefacts with this approach, a common approach in medical tomography (Zeng, 2010), but increases the number of samples collected.

In November 2021, a 20m x 20m grid, subdivided into 25 4m x 4m cells and oriented in the cardinal directions, was established in a flat, open lawn (Virginia Tech Drillfield). Aliquots of five compounds (α-pinene, adamantane, isoborneol, decane, and dodecane) were placed in watch glasses and distributed randomly in the grid, with each sample's coordinates noted. Ten samplers were placed at the southern and eastern edges of the grid (five per edge), centred in each 4m x 4m cell. A control was placed outside of the grid, 7m due north from the grid's NE corner. The transects covered two angles, both in

cardinal directions. The samplers were manually moved S to N and E to W at a rate of 1m every eight minutes. When the entire grid was traversed, the directions were reversed, and the grid recrossed at a faster rate of 1m every four minutes. The total sampling time was about four hours (233 minutes). The samplers used Tenax® TA cartridges and had flow rates of 250 ccm. A portable weather station (Ambient Weather WS-2000) recorded wind direction and wind speed with 5-minute time resolution. Since the control turned out to be downwind of the sampling grid, no correction was made to the cells' values.

**3 Results and Discussion**

**3.1 Sampler reproducibility**

During sampler reproducibility tests, two sets of samples were collected in an indoor environment. Thirty-one compounds were detected by both sets of samplers and quantified to ensure reproducibility and detect any differences in concentration (Fig. 2). Within each set of samplers, more than 85% of the quantified analytes agree to within 10% between samplers and

almost all other analytes agree within 20% (Fig. 2 inset histograms). This level of variability between samples is estimated as the approximate precision of chromatographic integration (Isaacman-VanWertz et al., 2017), suggesting samplers agree to within other measurement errors and produce highly reproducible samples.

Interestingly, although the two sets were only separated by about two meters, many of the analytes differ in their observed signal between the two sampler sets. In many cases, differences between sets are greater than the 10-20% difference between

samplers, suggesting a true difference in concentration. A two-tailed t-test at 95% confidence found that approximately two-thirds (20 of the 31 compounds) had statistically different concentrations between the sets (Table S1), demonstrating that fine-scale spatial differences exist and can be observed within the indoor environment. These differences are likely due to variability in proximity of emissions sources or perhaps small-scale discrepancies in air circulation. For example, Set A was located on the lab bench located closer to a Scanning Mobility Particle Sizer that uses 1-butanol, and it showed over 50%

higher levels of this compound relative to Set B. These data suggest that thoughtful placement of the sampling boxes can detect analyte concentration differences over small spatial distances and may be useful in locating point sources. Given that intragroup differences between samplers are reliably <15% (usually <10%), even small differences in concentrations can be measured.

## 3.2 Efficacy of ozone removal

Ozone removal was tested when sampling both indoor and outdoor air by comparing concentrations of reactive analytes both with and without ozone scrubbing. The ratio of analyte signal in the unscrubbed (i.e., ozone-exposed) sample to the scrubbed sample is interpreted as the reacted fraction of analyte upon exposure to the ozone in the sample flow. Higher reacted fractions are observed between the scrubbed and unscrubbed in the case of outdoor air due to higher expected ozone levels relative to the unoccupied and dark indoor environment (Fig. 3). Reacted fraction of each analyte is observed to correlate

with ozone reactivity. For the monoterpene analytes, the reacted fraction for outdoor air was significantly lower than for indoor air in all cases, though sulcatone ($k_{O3}$ = $2.6\times10^{-16}$ $cm^3$ molecule$^{-1}$ s$^{-1}$) was not observed in the indoor samples. BTEX compounds have comparatively negligible reactivity with ozone and show no significant differences with and without the ozone scrubber, demonstrating that analyte loss is not due to some confounding loss process (e.g., adsorption to the filter). Reacted fraction is approximately consistent with expected analyte loss as a function of reaction rate, assuming exposure

throughout the sampling time (8.7 hours for outdoor and 13.5 hours for indoor) to constant representative ozone concentrations. For this model, indoor conditions were assumed to be 5 ppbv  (Nazaroff and Weschler, 2022) and outdoor conditions assumed 40 ppbv, the approximate observed daily outdoor average during the sampling period. More information and statistical data on the selected analytes are given in Tables S2 and S3.

    This ozone removal approach has the additional benefit of removal particle-phase compounds that may not be removed by

the analytical system and could decrease the lifetime of the sampling cartridge. However, it has also been shown to have the downside of potentially removing some oxygenated gases, particularly those with lower volatility (Ngo et al., 2020; Pollmann et al., 2005). Estimates of thiosulphate scrubber lifetimes have been made and effects of humidity also have been studied; these studies observed lifetimes of 14 days at moderate flow rates (200 ccm) and ozone levels (50 ppbv),with longer lifetimes and more efficient scrubbing occurring at higher humidity (80% RH) (Ernle et al., 2023). It should be noted, for

this and all other sampler validation and field tests, longer sampling times and higher flow rates were chosen to stress test the components, but experimenters should be aware that such conditions may lead to breakthrough of the most volatile compounds; such breakthrough may explain the large error bars on benzene, the most volatile BTEX compound, in Fig.3. Nevertheless, possible breakthroughs would not affect the overall conclusions of this work.

    The qualitative agreement between this simplified model for analyte loss and observations supports the conclusions that

observed differences with and without the scrubber are attributable to removal of ozone, that the simple ozone scrubber employed is effective, and that a functioning ozone scrubber is necessary. Overall, the addition of a chemically reactive filter upstream such as the ozone scrubber here introduces the possibility of negative artifacts (removal of oxygenates) or positive artifacts (volatilization of particle-phase compounds), but as demonstrated by Fig. 3 is a necessary component to avoid removal of reactive gases. Previous work has explored other ozone removal approaches, but each comes with its own trade-

offs (World Meteorological Organization, 2023). Balancing the need for ozone removal with its disadvantages is a necessary aspect of sampling for reactive gases and should be considered in any interpretation of data from a given sample collection.

### 3.3 Measurements of temporal variability

Given the high precision observed in samples (Section 3.1), temporal variability in real ambient concentrations is theoretically measurable with these samplers, which we demonstrate in practice here. Concentrations of a set of 13 analytes commonly observed in indoor environments were monitored by a set of three samplers in an indoor atmosphere over 3 weeks, with five representative compounds shown in Fig. 4. For most analytes, compounds are observed to vary by approximately a factor of three around the mean, significantly larger than uncertainty in the measurement (estimated as the standard deviation of the three samples in each set). This variation is likely due to variation in the operation of the building's HVAC system, periodic cleaning of the unoccupied room by the housekeeping staff, and the occupancy and use of adjacent rooms. Additional analytes not included in the plot showed similar trends with respect to day-to-day variability and error (Table S4). Overall, average variability between samplers is less than 10%, while measured concentrations have a standard deviation of 50%. Day-to-day differences also suggest that some compounds that have previously been observed as artefacts from sorbent decomposition (e.g., toluene (Cao and Nicholas Hewitt, 1994; MacLeod and Ames, 1986)) likely represent real ambient components in this environment; artefacts would be expected to have higher variability between samplers. These data demonstrate that sampler precision remains high when a lower number of replicates is used, allowing these systems to capture real temporal variability on a scale of hours or days. An alternative approach to detecting temporal variability requiring less operator interaction would deploy multiple samplers at one location programmed for sequential sampling using staggered start times.

### 3.4 Mapping spatial variability

A major improvement of programmable, low-cost, portable samplers over other sampling tools is the possibility of a distributed simultaneous collection network of multiple samples to generate a map of analytes of interest. To demonstrate and evaluate this capability, known emissions sources were dispersed throughout a field and sampled. As described in Section 2.4.2, sampling was not conducted by placing each sampler in evenly spaced grid cells, but rather using orthogonal transects across a region of interest in which emissions sources were placed (Fig. 6a). This novel "transect mapping" approach enables higher resolution than the number of samplers. Since the goal is to identify emissions sources, concentrations in each cell are estimated as the minimum concentration of the two transects that cross each grid cell. This approach does not necessarily capture fine concentration gradients, but rather identifies hotspots of concentration, which are expected to have high concentrations in both transecting samples and thus a high minimum concentration. Estimated concentrations in each grid cell are indeed found to be highest near the known emissions sources (Fig. 6c-g). The maximum transect-averaged concentrations of analytes ranged from 2 pptv (isoborneol) and 2,600 pptv (decane).

Concentrations of the three analytes with only a single emissions source (α-pinene, adamantane, and isoborneol, Fig. 6e-g) are highest just northeast of where the aliquot was placed. This effect is expected given the prevailing moderate

southwesterly winds during the sampling period (Fig. 6b). All grid cells were observed to have some α-pinene, likely due to its presence as a ubiquitous ambient gas, but estimated concentration was substantially greater downwind of the emission source. In contrast, adamantane and isoborneol are not expected to be present in high concentrations under ambient conditions and neither are  observed in the majority of upwind grid cells. Adamantane and isoborneol also have far lower vapor pressures (2.99 Pa and 0.06 Pa, respectively) compared to the other analytes, leading to low emissions and low concentrations (pptv-level) that are more likely to fall below limits of detection in cells not influenced by emissions.

For analytes with two emissions sources (decane and dodecane, Fig. 6c-d), the results are a bit more complicated and taking the minimum value for the two crossing samplers generates some artefacts. For example, the southern emission source of decane appears to have some concentration in the cell due west of the source, though this is unlikely as it is upwind of both emission sources. That cell is a crossing point of two transects containing sources, so taking the minimum still gives a likely overestimate, demonstrating a phantom image effect that can be caused by back projection with only two orthogonal angles (Zeng, 2010). Despite the phantom images, the low concentration expected in the southwestern corner and western edge of the grid is observed. Some artefacts are consequently possible when multiple sources are present, yet this approach nevertheless reasonably narrows the location of emission sources even in complex cases in addition to performing well in single-source cases.

Transect mapping provides a novel approach to industrial or field applications in which a point source needs to be located, as it provides an approximate source location with higher resolution than can be achieved by the same number of samples in a stationary grid. While stationary sampling across a grid of $n$ cells would require $n$ samples, transect mapping with the same number of samples using a square grid can generate a grid with $\frac{1}{4}n^2$ equally sized square grid cells. For example, in the experimental grid shown in Fig. 6, with five transects in each direction, 10 samples were collected to yield a grid of 25 grid cells. The advantages of the transect mapping approach increase with larger sampling areas. Because the area of the sampling grid increases as the square of a side length, for stationary sampling, doubling a side length requires a fourfold increase in the number of samples achieve grid cells of the same size. In contrast, in the transect mapping approach, doubling the side length requires only a doubling of sample number. There is also substantial opportunity for potential optimization of the transect mapping approach. Adding additional angles would reduce artefacts and phantom images due to multiple sources, but at the cost of more samples to achieve the same resolution. For example, transects in three evenly spaced angles generates a hexagonal grid of approximately $\frac{1}{6}n^2$ equally sized triangular grid cells; achieving a resolution similar to that shown in Fig. 5 requires 12 samples (fourtransects in each of three directions, 24 grid cells; Fig. S1), but would not generate phantom images in the case of multiple emissions sources. A detailed discussion is included in Section S4 and investigators will need to consider these trade-offs in the experimental protocols.

### 3.5 Limits of detection (LOD)

Limits of detection (LODs) for the samplers follow typical patterns of gas sampling cartridges, depending almost entirely on the levels of artefacts from sorbent decomposition, the sensitivity of the detector, and the sampling duration. In theory, the investigator could measure a very low-abundance (e.g., ppqv) analyte of interest simply by running the samplers for long periods. In practice, the sorbent binding sites may become saturated with other, more abundant, species before the desired compound can be collected in sufficient quantity, and/or higher volatility analytes may break through the sampling cartridge.

We estimate an LOD for the specific sampling and analytical system used here, using commercially prepared cartridges analysed by GC/MS. We consider two very different analytes, disparlure and α-pinene. Disparlure, a 19-carbon epoxide, is a spongy moth (*Lymantria dispar* (L.)) sex pheromone occurring in very low concentrations, whilst α-pinene, a 10-carbon hydrocarbon, is an abundant monoterpene commonly emitted from conifers and often present in fragranced consumer products. For disparlure, 0.76 ng of analyte "on column" (the mass on the sorbent tube, either from sampled air or injected as

a standard onto the sorbent and thermally desorbed into the GC-MS) was found to provide a signal-to-noise ratio, S/N = 3. In contrast, α-pinene, a less polar compound and thus more conducive to GC analysis, yielded S/N=3 with approximately 0.1 ng on-column. Assuming only one hour of sampling at the approximate maximum flow rate of 250 ccm (250 ccm was consistently attainable by all samplers in all experiments), limits of detection are roughly 6 pptv and 1 pptv for disparlure and α-pinene respectively. These estimates are comparable to those of other researchers for commercially prepared

cartridges, though other work has found that custom-prepared cartridges can significantly improve backgrounds and lower LOD (Sheu et al., 2018). LOD may be higher for compounds that exhibit significant background contamination or artefacts from a given sorbent.

### 4 Conclusions and Applications

The portable VOC samplers described in this work offer the researcher great flexibility since the samplers are portable,

robust, and straightforward to operate. We have demonstrated high levels of precision between samplers, while at the same time showing that the samplers can record significant differences in analyte concentration over small spatial scales. The samplers can be fitted with ozone scrubbers to preclude loss of vulnerable compounds to ozonolysis, while at the same time leaving unsusceptible analytes unaffected. A time series was performed with a small number of the samplers and showed the ability to distinguish temporal (day-to-day) variation in analyte levels while maintaining small standard deviations for a

given day. The transect mapping exercise was promising for determining source allocation, though sampling grids containing analytes with more than one point source may require transects at three or more angles and/or a more sophisticated way of estimating compound concentrations. These results suggest that our VOC samplers can fill a niche in measurements of atmospheric organics and may be ideal for biologists doing field studies in remote locations or monitoring pollutants of interest in industrial settings.

*Author Contributions.* **James Hurley**: Validation, Investigation, Data analysis and Writing. **Alejandra Caceres**: Methodology (development of portable sampler prototype and construction/maintenance of the samplers). **Deborah McGlynn**: Methodology (development of the portable sampler prototype). **Mary Tovillo**: Methodology (development of the portable sampler prototype). **Suzanne Pinar**: Investigation and Data analysis (transect mapping concept). **Roger Schürch**: Conceptualization and Methodology (transect mapping concept). **Ksenia Onufrieva**: Conceptualization, Methodology, Funding acquisition, Investigation. **Gabriel Isaacman-Van Wertz**: Supervision, Conceptualization, Methodology, Writing-Review and Editing.

*Competing Interests.* The authors declare that they have no conflicts of interest.

*Acknowledgements/Financial Support.* This research was supported in part by the National Science Foundation (AGS 1837882 and AGS 2046367) and the Virginia Tech CALS Strategic Plan Advancement program. Alejandra Caceres and Mary Tovillo were supported in part by the Virginia Tech Multicultural Academic Opportunities Program. Roger Schürch was supported by the USDA National Institute of Food and Agriculture, Hatch Project VA-160129. Thanks to Alice Isaacman for assistance in determining the relationship between number of samples and resolution in the case of triangular grid cells.

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

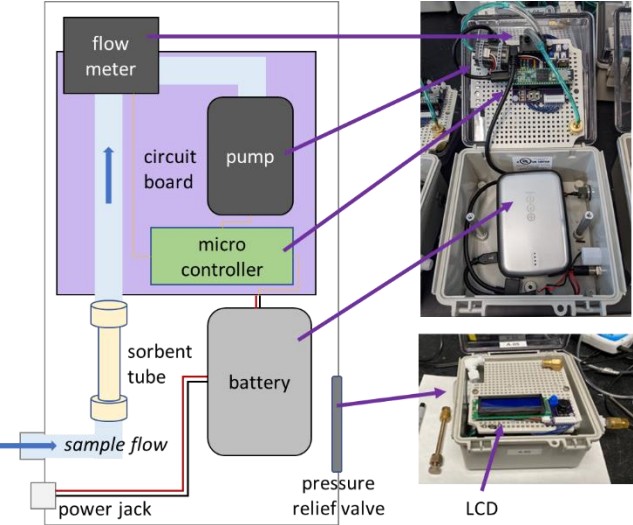

**Figure 1: Simplified schematic (left) and photos (right) of sampler design, with key components labelled.**


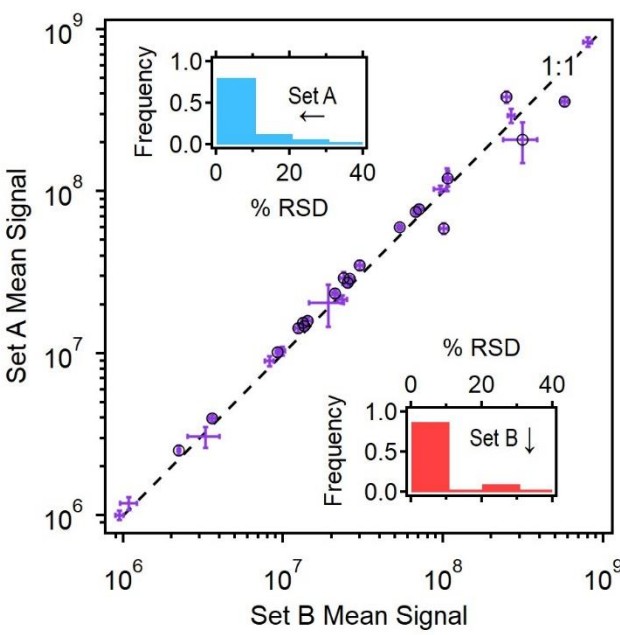

**Figure 2: Mean signal of 31 analytes measured in two sets of samplers (Sets A, n=5, and Set B, n=7) spaced approximately two meters apart. Standard deviation shown as error bars, with black circles indicating the 20 analytes found to be statistically significantly different between the two sets (t-test, 95% confidence). Insets show frequency distributions of % relative standard deviation for each set (Set A in blue, Set B in pink).**


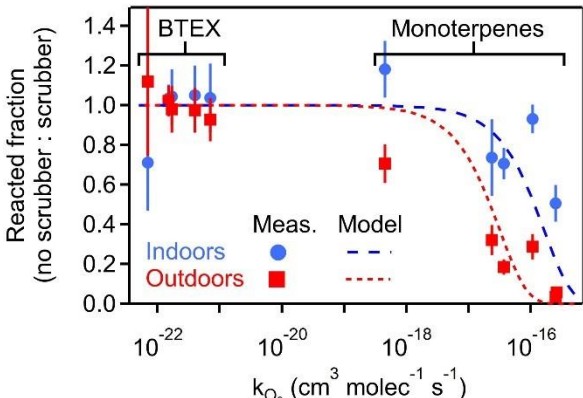

**Figure 3: Fraction of reactive (monoterpene) and non-reactive (BTEX) analytes reacted on the sampling media when sampled without ozone removal, quantified as the ratio of signal when sampled without vs. with an ozone scrubber. Shown as a function of the ozone reaction rate coefficient ($k_{O3}$), with vertical bars for propagated error in the ratio. Model analyte loss shown as dashed lines representing outdoor sampling (8.7 hours, 40 ppbv) and indoor sampling (13.5 hours, 5 ppbv). Analytes and rate coefficients are given in Supplemental Table S2.**

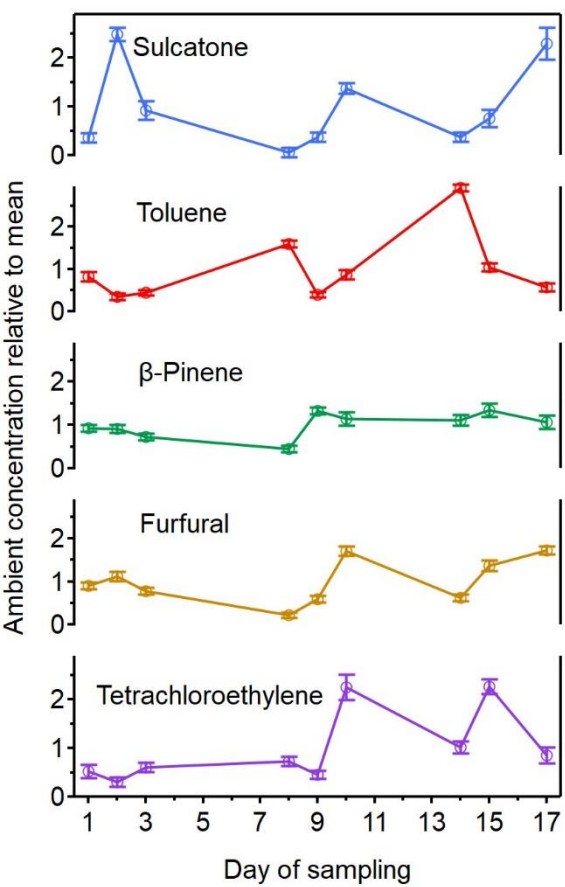


**Figure 4: Nine normalized values (daily measurement/aggregate mean) plotted against day for five (of 13) selected analytes. Samples were taken in spring in an indoor room containing plants arranged on a greenwall. Error bars are the standard deviation of the daily triplicate measurements.**


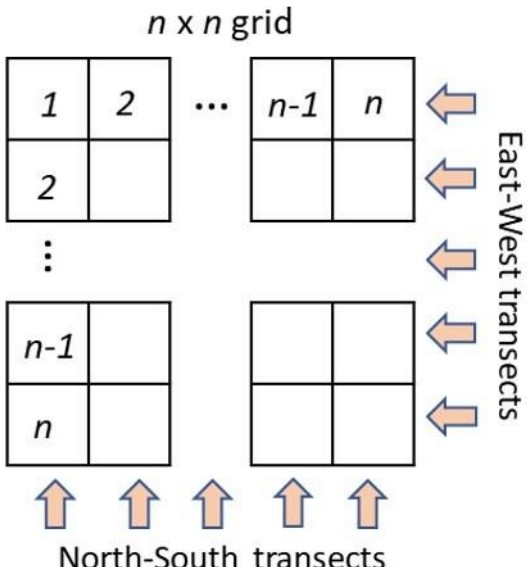

**Figure 5: A generalized sampling schematic showing an n x n grid with sampler movement marked by the arrows and oriented in the cardinal directions.**

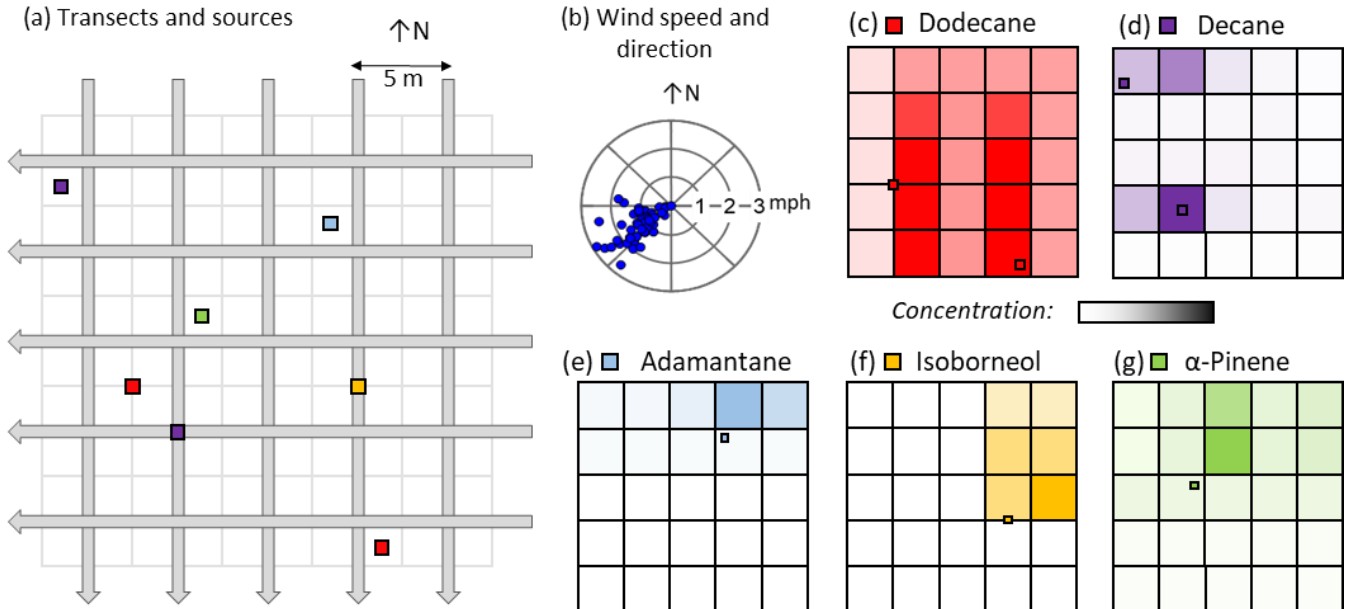

**Figure 6: Overview of transect mapping.** Set up shown as (a) N-S and E-W transects sampled through a 5 x 5 grid of 5m x 5m grid cells with emission sources are shown as coloured squares, and (b) wind direction and speed measured during sampling. Results are shown as colour scales representing the minimum of the two transect-average concentration observed for each cell, with colour scales normalized between zero (no fill) and the maximum observed transect-average concentration for the compound (dark fill). Compounds are (c) dodecane, red, maximum concentration of 50 pptv; (d) decane, purple, maximum concentration of 2,600 pptv; (e) adamantane, blue, maximum concentration of 7 pptv, (f) isoborneol, yellow, maximum concentration of 2 pptv, and (g) α-pinene, green, maximum concentration of 400 pptv. For each result grid, the emission location of the analyte is shown.