# Peer review of "Portable, low-cost samplers for distributed sampling of atmospheric gases"

_EGUsphere, 2023_

## Author Comment (AC1)

**Response to Referees**

We thank the reviewers for their careful reading of our manuscript and are pleased with their assessment that this work is promising. We appreciate the concerns raised by the reviewers, and hope they agree that our revisions of this manuscript satisfactorily address those concerns.

Please find below responses to all individual comments. Reviewers' comments are in blue, our response is in black, and any new text is *italicized.*

**Anonymous Referee #1**

This manuscript describes the development of a portable, low-cost adsorbent tube sampler for field deployment. The authors describe their testing of sampler reproducibility and ozone scrubbing, as well as their deployments to investigate long-term temporal variability and spatial variability measurement capabilities. The programmable nature of this sampler along with its small size make it an attractive tool for researchers or industry professionals to monitor air quality. I particularly enjoyed reading about the transect mapping approach the authors discussed. I do think the sampler's applications are somewhat limited – e.g., can only sample on one cartridge at a time before needing to be replenished, still has complex data analysis and thus requires operator expertise – but the study represents useful developments in lower cost sampling strategies that preserve a high level of chemical detail. The manuscript was very well written and the figures were clearly presented. I recommend this paper for publication in AMT after minor revisions.

We thank the Reviewer for taking the time to review this manuscript and offer their suggestions. Please find our responses below.

**Introduction:** very nicely written. A few typographical comments:

1a) Line 40: suggest revising to "does damage" instead of "doing damage"

We have revised the text as follows (Introduction, line 39-40):

"…while ozone is a powerful oxidant that adversely affects humans and *does damage* to crops, forests…"

1b )Line 68: "reduce" instead of "reduces"

We have revised the text as follows (Introduction, line 68):

"…for offsite analysis to *reduce* the need for complex instrumentation…"

1c) Line 83: Typically these adsorbent cartridges do need to be cleaned (e.g., via thermal conditioning)

We concur that sorbent cartridges that are new or have been stored for a long time need to be conditioned  but note that cleaning between regular uses is achieved simply by analyzing the previous sample. EPA Method TO-17 (https://www.epa.gov/sites/default/files/2019-11/documents/to-17r.pdf), dealing with sorbent tube sampling of VOCs, calls for conditioning newly packed (or purchased) sorbent

tubes but does not require cleaning between uses. See specifically sections 6.2.3: "On second and subsequent uses, the tubes will generally not require further conditioning" and 6.15.1: "When tube analysis is completed, remove the tubes from the thermal desorber and…. re-store the tubes in a cool environment (<4°C) until the next use."

We found that cartridges run on the thermal desorption system for two cycles (of TD-GC-MS) showed second chromatograms indistinguishable from the accompanying blank tubes. Avoiding cleaning between samples saves time and increases the lifetime of the sorbent tubes. Although we often used recently analyzed tubes for another round of sampling without conditioning, tubes that had not been used for more than a few days were thermally conditioned to ensure their cleanliness. We leave it to the investigators to decide when conditioning their sorbent tubes is warranted.

We have added a reference to the EPA method and revised the text as follows (Introduction, lines 84-86):

"Advantages of sorbent tubes and TD over whole air samplers are the tubes' small size and portability, lower cost, increased ability to transfer less volatile ($C_{10}$ and greater) and polar compounds, *and greater ease of cleaning*."

We have also explained this aspect of our protocol to the Methods section (2.2, lines 139-144)

"Cartridges were thermally desorbed for analysis (TD-100-xr, Markes International) by gas chromatography and mass spectrometry (GC/MS; Focus GC/DSQ II MS, Thermo Scientific). *These sorbents target broad ranges of mid- and lower-volatility gases so are used to validate the samplers across a wide range of gases, including the more moderate-volatility gases that are more likely to suffer sampling losses on surfaces*. Tubes were thermally desorbed at 250°C for 8 minutes, with a desorption flow of 50 mL/min and a flow path temperature of 190°C. *Due to nearly constant use of the sorbent cartridges, it was determined that analysis of the tubes was equivalent to a cleaning step; however, tubes that had not been analysed for more than a few days were conditioned before use.*"

**1d) Line 89: I'd also consider re-emphasizing the need for relatively complex data analysis with adsorbent sampling**

We agree with the reviewer that a drawback of sorbent tubes is the need for an operator able to run the analytical instrumentation and analyze the collected chromatograms. Our portable samplers are advantageous over other sampling options in that a researcher can utilize the aforementioned features (programmability, portability, etc.) to collect samples across spatial or temporal domains. Unfortunately, there are few analytical tools available for real-time, in-field VOC analysis at ambient concentrations; those that exist are generally limited to a single point in space or time and require operator expertise.

We add the qualification in the subsequent paragraph, altering the text as follows (Introduction, lines 96-105):

"We present here a versatile, lightweight, battery- and solar-powered sampler capable of collecting atmospheric VOCs on sorbent tubes for GC-MS analysis, with simultaneous sampling across a network of samplers enabled by programmable sampling and weatherproof design. By enabling coordinated and self-powered sampling across multiple locations, these samplers fill a current gap in the need for affordable, lightweight devices capable of capturing spatial and temporal variabilities *(on scales as small as meters and as fast as ~hours)* of analytes of interest in diverse settings. *As will all analyses relying on sorbent cartridges or chromatography, sample analysis does require skilled personnel to operate the instrumentation and process the data, though the lower costs and mobility of the samplers should offer researchers much more latitude in sampling protocols.* In this work, we describe the design of the sampler, demonstrate the reproducibility of sampling, and demonstrate the utility of a simultaneous network of multiple samplers, including a unique approach to pinpointing an emission source across a spatially heterogeneous area."

1e) Line 93: Would you say that you can capture temporal variability adequately with the hours-long sampling here? Certainly you can to some extent, but I'd consider qualifying what you mean by temporal variability somewhere. When I think of temporal variability, I really think of online VOC measurements like PTR-MS. For clarity, it would help to mention "over hours-long periods" or similar.

We agree with the reviewer that the temporal variability feasibly measured by sampling and offline strategies is much longer-term than the fast time response of instruments like a PTR (though of course GC based methods have other strengths). Sampling over an hour or a few hours is typical for cartridge-based approaches; this can be achieved here using multiple samplers with staggered programmed start times, operators switching tubes for intensive sampling, or a modified multi-channel sampler (which we have built and tested but is not a focus of the present work).

We have changed the text in two places to clarify the statement of temporal variability.

(Introduction, lines 98-100)

"By enabling coordinated and self-powered sampling across multiple locations, these samplers fill a current gap in the need for affordable, lightweight devices capable of capturing spatial and temporal variabilities (*on scales as small as meters and as fast as ~hours*)  of analytes of interest in diverse settings."

(Section 3.3, lines 300-304)

"These data demonstrate that sampler precision remains high when a lower number of replicates is used, allowing these systems to capture real temporal variability *on a scale of hours or days. An alternative approach to detecting temporal variability requiring less operator interaction would deploy multiple samplers at one location programmed for sequential sampling using staggered start times.*"

**Materials and methods:**

1f) I appreciate the low operator involvement associated with your design (i.e., pre-programmable, very few controls to navigate), but since it can only sample one cartridge at a time, the operator involvement actually seems fairly high if cartridges need to be continually replaced. Would you consider building a multi-cartridge system in the future? While that might come with its own challenges, it would enhance your instruments capabilities significantly. I'd suggest mentioning this somewhere as a limitation of your proposed system.

We agree with the reviewer that the operational costs of offline sampling are high. We have built and tested a multichannel sampler prototype that allows switching sequentially though eight sampling cartridges and uses many of the same components as the single-port VOC samplers covered in the manuscript. The multichannel sampler was deployed in a pilot study and detected the anticipated diurnal patterns of monoterpenes and BTEX compounds. It was not included in this work as the design and applications are a bit different, so it is difficult to include in this manuscript's scope; furthermore, temporal variability can be captured with these single-port samplers by deploying multiple samplers to a single location. Multiple single-channel samplers described here could be programmed for sequential sampling, enabling temporal variability to be captured with the need to only interact with the samplers once per day.

We have added a comment in the text about the development of the multichannel samplers in Section 2.1 (lines 129-134)

"The sampler dimensions are approximately 14 cm x 22 cm x 10 cm and weigh 1 kg. Most of the size and weight of the sampler is due to the ruggedized box and high-capacity battery; the electronic components are all mounted on a panel within the case that can be removed to create a 250 g version that is suitable and has been tested for short-term deployment on small mobile platforms such as drones. *In addition, many of the components used in this work are suitable for the development of a multi-channel sampler prototype, which was built and tested to allow sequential sampling but is outside the scope of this manuscript, which focuses on distributed and portable sampling.*"

1g) Does the temperature in the box increase at all with the operation of the pump and other electronics, and what does that do for the retention of volatiles in the cartridge? Do you have any sort of temperature monitoring or temperature tests to address this? If so I suggest including those results or commenting on what is expected. Without an on-board fan or anything, I am concerned that you are seeing significant temperature increases in your sampler box especially when deployed in the field in the sun.

In the original manuscript, it was not clear that there is a temperature sensor on the circuit board to monitor temperature in the box. In our testing, there has been no indication that the temperature inside the box changes more than 1°C and we do not consider this to be an issue. The design of the boxes helps alleviate this potential problem because the sampled air is exhausted by the pump into the interior of the box and exits the pressure release valve as depicted in Figure 1. We estimate the unoccupied volume of the box is ~1 L, so a sample flow rate of 250 ccm flushes the box every few minutes and the temperature inside the box is maintained near ambient temperature. While an active fan could be added, this would decrease the battery life of the samplers. There is no indication it is necessary, and it would still maintain the box only at ambient temperature (not below ambient).

The most volatile analytes may show some losses at high temperatures and the user should be aware of this possibility and perhaps try to position the boxes in shade. This issue is inherent in any cartridge sampling, and it is always necessary for a researcher to select an adsorbent that is capable of quantitative collection at expected sampling temperatures.

We have altered the text to make the temperature measurement clear and discuss this issue:

Section 2.1, lines 117-119:

"Communication between the pump, flowmeter, and microcontroller is managed by a custom circuit board *with an onboard temperature sensor (Microchip MCP9700)*, with firmware written in the Arduino IDE."

Section 2.1, lines 125-129:

"The case and external components are all at least IP66-rated, ensuring the samplers are rugged and weatherproof; successful deployment in this work occurred under both wet and freezing cold conditions. *Because sampled air is exhausted into the box and out through the pressure release valve (Fig. 1), residence time of air in the box is only a few minutes and the internal box temperature is maintained near ambient temperatures*."

1h) Relatedly, did you perform any breakthrough testing at such long sampling times (e.g., the many hours described in the ozone scrubber testing, the roughly 1-day long samples described in the long-term temporal variability section)? I'd like to see some additional discussion of sorbent breakthrough as well as possible sorbent breakthrough under warmer temperature conditions in the box (as mentioned above).

Breakthrough of analytes through sorbent tubes is a concern and has been the subject of a fair amount of previous work and guidance provided by manufacturers and agencies such as the EPA. We did not perform specific breakthrough testing in this work, as the primary focus is on the sampling strategy and device, not quantitative measurements of any specific analyte to address any specific scientific question.

Approximate sample volumes for tests described in this work were: 60 liters for transect mapping (4 hours at 250 sccm), 110 liters for reproducibility tests (7.5 hours at 250 sccm), 150 to 250 liters for ozone removal tests (8.7 and 13.5 hours at 300 sccm) and 360 liters for indoor day-to-day variability measurements (24-hour sampling at 250 sccm). These volumes are certainly on the high end for typical sorbent sampling. For many of the analytes reported, breakthrough is not expected to be a significant issue based on previously reported breakthrough volumes of hundreds of liters; exceptions are the lightest aromatics like benzene and toluene, which may suffer some breakthrough. Resources on breakthrough volumes are provided by manufacturers:

https://www.sigmaaldrich.com/deepweb/assets/sigmaaldrich/marketing/global/documents/103/692/t402025.pdf

https://www.sisweb.com/index/referenc/tenaxta.htm

Our goal was to examine the performance of the sampler boxes rather than make measurements of absolute concentrations. Long sampling times and high flow rates were selected to operate the components of the samplers under rigorous conditions and test battery life (i.e., stress test). For long sampling times, users could operate instead at the minimum 50 sccm flow rate to reduce sample volume and should make careful choices of adsorbents. Breakthrough may occur for some analytes in the longest tests, but for the most part there are no major observed differences in the reproducibility of these compounds; this issue may be responsible for the large error bars on benzene in the ozone removal tests (Fig. 3).

We agree that breakthrough volumes are an important consideration and have modified the text accordingly in two places:

 (Introduction, lines 87-91)

"Drawbacks of the tubes are the limited ability to sample the most volatile compounds ($<C_4$), the presence of artefacts, the need to refrigerate unanalysed samples, and the requirement for a sampling pump (Betz and Supina, 1989; Bianchi and Varney, 1993; Ciccioli et al., 1992; Woolfenden, 1997). *Furthermore, accurate measurements of absolute analyte concentrations require knowledge of breakthrough volumes, i.e. the volume of sampled air that would completely elute the analyte through the sorbent and lead to underestimates (Harper, 1993)."*

(Section 3.2, lines 274-278)

*"It should be noted, for this and all other sampler validation and field tests, longer sampling times and higher flow rates were chosen to stress test the components, but experimenters should be aware that such conditions may lead to breakthrough of the most volatile compounds; such breakthrough may explain the large error bars on benzene, the most volatile BTEX compound, in Fig.3. Nevertheless, possible breakthroughs would not affect the overall conclusions of this work."*

1i) Line 195-200: this description is challenging to understand on its own but much clearer once the reader looks at Figure 5, can you add a reference to figure 5 somewhere in this discussion to draw attention to it? The figure is great and helped clarify the text significantly.

We recognize that the text can be a bit confusing and welcome the reviewer's suggestion. We have added explanatory text and added a figure, Figure 5, to Section 2.4.2 to better familiarize the reader with our sampling scheme before describing our actual experiment in Figure 6 (formerly Figure 5).

(Section 2.4.2, lines 212-216)

"To examine the utility of distributed sampling to locate emissions sources, the target area was split into a grid of *n* x *n* cells. *Fig. 5 shows a generalized sampling grid with n x n cells and the cardinal directions forming the two sampling angles of 0° and 90°. A three-angle grid would have sampling angles of 0°, 60°, and 120°; a four-angle grid would have angles of 0°, 45°, 90°, 135°, and so on. See Section S4 for further discussion of the number of angles and optimization of the transect mapping approach.*"

[Figure]

**Figure 5: A generalized sampling schematic showing an n x n grid with sampler movement marked by the arrows and oriented in the cardinal directions.**

**Results:**

 You describe possible analyte loss to the ozone scrubbing filter here, and mention that species with low ozone reactivity had minimal concentration difference with and without the scrubber, ruling out adsorption to the filter. How would you expect more functionalized gases to behave here? I would not anticipate that BTEX species would be especially susceptible to filter-related losses but I am concerned about losses of other oxygenated VOCs for example. Can you please comment on this possibility? Or provide some discussion of the limitations of this testing with high volatility non-functionalized species?

Loss of oxygenated gases can be an issue with ozone scrubbers.

These ozone scrubbers are based on the work of Pollman et al., 2005, (https://pubs.acs.org/doi/10.1021/es050440w), who investigated ozone scrubber effects on sesquiterpenes, aromatics (with higher masses than the BTEX compounds), a biogenic ketone (geranylacetone) and oxygenated sesquiterpenes (*cis-* and *trans*-nerolidol). High filter losses were observed for the two oxygenated sesquiterpenes (see Pollman's Figure 3 and Supplemental Table 2). *Cis-* and *trans*-Nerolidol have a hydroxy group than may be interacting strongly with the glass filter (Ngo et al., 2020, doi:10.1038/s41529-019-0105-2). Our only oxygenated compound was sulcatone, a ketone with a double bond, detected only outdoors. Sulcatone's unscrubbed/scrubbed ratio corresponds well to its $k_{O3}$ value, and like the ketone in Pollman's experiments (much lower volatility geranylacetone) we think its filter losses should be minimal. However, some losses of oxygenates are likely. Unfortunately, we are always balancing loss of hydrocarbons due to ozone and loss of oxygenates due to removal of ozone.

We have modified the text as follows:

(Section 3.2, lines 269-274):

"*This ozone removal approach has the additional benefit of removal particle-phase compounds that may not be removed by the analytical system and could decrease the lifetime of the sampling cartridge. However, it has also been shown to have the downside of potentially removing some oxygenated gases, particularly those with lower volatility (Ngo et al., 2020; Pollmann et al., 2005). Estimates of thiosulphate scrubber lifetimes have been made and effects of humidity also have been studied; these studies observed lifetimes of 14 days at moderate flow rates (200 ccm) and ozone levels (50 ppbv),with longer lifetimes and more efficient scrubbing occurring at higher humidity (80% RH) (Ernle et al., 2023).* "

1k) For field sampling, have you considered the impacts of water uptake on your sodium thiosulfate filter? How would that impact ozone scrubbing over long sampling times? Do you expect any reactivity of particle-phase species on the surface of your filter that might result in the volatilization of gases that you collect in the adsorbent tube? I suggest adding some comments about necessary considerations when deploying a glass fiber filter upstream of the adsorbent cartridge.

As noted by the Reviewer here and above, the inclusion of an ozone scrubber, while necessary, adds some potential complications. Ernle et al., 2022 ( https://doi.org/10.5194/amt-16-1179-2023 ) investigated the effects of humidity and found, perhaps counterintuitively, that both scrubber efficacy and lifetime increased under high humidities compared to dry conditions (see their Sections 3.2, 3.3). They explain this from the scrubber reaction (their R1):

R1      $2 S_2O_3^{2-} + O_3 + 2 H^+ \leftrightarrow S_4O_6^{2-} + O_2 + H_2O$

Adding water to the system pushes the equilibrium back to the left, regenerating thiosulphate from tetrathionate. See modifications to the text highlighted in the comment above regarding impacts of humidity and the detailed work of Ernle et al.. (Section 3.2, lines 272-274).

This style of ozone scrubber has been used for weeks-long deployment on other systems (including by our group, https://doi.org/10.5194/acp-21-15755-2021), so has been demonstrated as a viable long-term option. We agree with the reviewer that particle collection on the filter could lead to potential artifacts, such as volatilization of particle-phase compounds, though this is expected mostly for lower-volatility compounds and is an issue under any sampling regime. It is not clear how reactions on the filter could dislodge adsorbed species from the sorbent, though they could generate volatile compounds that are subsequently captured. We highlight the upsides and downsides of this approach in the revised manuscript:

Section 3.2, Lines 279-286:

"The qualitative agreement between this simplified model for analyte loss and observations supports the conclusions that observed differences with and without the scrubber are attributable to removal of ozone, that the simple ozone scrubber employed is effective, and that a functioning ozone scrubber is necessary. *Overall, the addition of a chemically reactive filter upstream such as the ozone scrubber here introduces the possibility of negative artifacts (removal of oxygenates) or positive artifacts (volatilization of particle-phase compounds), but as demonstrated by Figure 3 is a necessary component to avoid removal of reactive gases. Previous work has explored other ozone removal approaches, but each comes with its own tradeoffs (World Meteorological Organization, 2023). Balancing the need for ozone removal with its disadvantages is a necessary aspect of sampling for reactive gases and should be considered in any interpretation of data from a given sample collection.*"

1l) Figure 3: would be helpful to make BTEX and Monoterpenes different shapes.

We think it is clearer to keep the two sampling environments separated by shapes so that color is not the only distinguishing feature, particularly because the two compound types are well separated along the x-axis. To address the concern raised by the reviewer, we have added brackets to Figure 3 to make clear which points represent each compound type.

[Figure]

**Figure 3: Fraction of reactive (monoterpene) and non-reactive (BTEX) analytes reacted on the sampling media when sampled without ozone removal, quantified as the ratio of signal when sampled without vs. with an ozone scrubber. Shown as a function of the ozone reaction rate coefficient ($k_{O_3}$), with vertical bars for propagated error in the ratio. Model analyte loss shown as dashed lines representing outdoor sampling (8.7 hours, 40 ppbv) and indoor sampling (13.5 hours, 5 ppbv). Analytes and rate coefficients are given in Supplemental Table S2.**

**Anonymous Referee #2**

This is a very nice piece of work demonstrating the development of a relatively inexpensive adsorbent cartage based portable VOC sampler for a field study. To ensure that multiple samplers can be used in a field study that examines temporal and spatial variability, the authors performed comprehensive sampler characterisation experiments that tested their stability, repeatability, and reproducibility. Furthermore, the authors showed a novel concept of transect mapping to pinpoint emission sources. Typically, a sampler needs to be located in each grid cell to pinpoint emission sources, but this makes such an exercise extremely costly. The authors demonstrated that their strategy allowed them to use much fewer samplers, for example, only 10 samplers to locate the emission sources from 25 grid cells. The manuscript is clear and concise and I recommend its publication once some minor technical issues are addressed.

 We thank the Reviewer for their time and suggestions. We address your specific comments below.

2a) Line 46: "hydroxyl" should be "hydroxyl radical".

Right. We have revised the text as follows (Introduction, lines 46):

"…promote the formation of oxidants (e.g., hydroxyl *radical*, ozone), the emission of…"

2b) Line 83: A TD adsorbent tube requires a post-use 'bake out' step prior to reuse.

This concern is raised by Referee #1 as well. We concur that sorbent cartridges that are new or have been stored for a long time need to be conditioned but note that cleaning between regular uses is achieved simply by analyzing the previous sample. EPA Method TO-17 (https://www.epa.gov/sites/default/files/2019-11/documents/to-17r.pdf), dealing with sorbent tube sampling of VOCs, calls for conditioning newly packed (or purchased) sorbent tubes but does not require cleaning between uses. See specifically sections 6.2.3: "On second and subsequent uses, the tubes will generally not require further conditioning" and 6.15.1: "When tube analysis is completed, remove the tubes from the thermal desorber and…. re-store the tubes in a cool environment (<4°C) until the next use."

We found that cartridges run on the thermal desorption system for two cycles (of TD-GC-MS) showed second chromatograms indistinguishable from the accompanying blank tubes. Avoiding cleaning between samples saves time and increases the lifetime of the sorbent tubes. Although we often used recently analyzed tubes for another round of sampling without conditioning, tubes that had not been used for more than a few days were thermally conditioned to ensure their cleanliness. We leave it to the investigators to decide when conditioning their sorbent tubes is warranted.

We have added a reference to the EPA method and revised the text as follows (Introduction, lines 84-86):

"Advantages of sorbent tubes and TD over whole air samplers are the tubes' small size and portability, lower cost, increased ability to transfer less volatile ($C_{10}$ and greater) and polar compounds, *and greater ease of cleaning*."

We have also explained this aspect of our protocol to the Methods section (Section 2.2, lines 139-144)

"Cartridges were thermally desorbed for analysis (TD-100-xr, Markes International) by gas chromatography and mass spectrometry (GC/MS; Focus GC/DSQ II MS, Thermo Scientific). *These sorbents target broad ranges of mid- and lower-volatility gases so are used to validate the samplers across a wide range of gases, including the more moderate-volatility gases that are more likely to suffer sampling losses on surfaces*. Tubes were thermally desorbed at 250°C for 8 minutes, with a desorption flow of 50 mL/min and a flow path temperature of 190°C. *Due to nearly constant use of the sorbent cartridges, it was determined that analysis of the tubes was equivalent to a cleaning step; however, tubes that had not been analysed for more than a few days were conditioned before use.*"

2c) Line 101: Have the authors considered an umbrella or a similar attachment at the end of the Teflon tube for rain protection? If so, please describe it here.

We have run the samplers in rainy conditions with no problems, but not in torrential rains, gale force winds, etc., so this is a point worth considering. Rather than an umbrella, which may affect advection/convection patterns and even cause movement of the sampler box in gusty conditions, we think it would be simpler to put a Teflon elbow connection in the sampling port and have the sampling inlet pointing towards the ground.

2d) Line 124: What are the authors' criteria for selecting a specific adsorbing polymer for different experimental condition or target VOCs? An additional description about the choice of adsorbing polymer will be good here.

A large body of literature exists on selecting an appropriate adsorbent to address a specific measurement need or scientific goal. In this work, our goal was simply to collect as many analytes as possible and see how the samplers performed (precision, required sampling times, etc.). Generally, lower volatility compounds are more likely to be lost to surfaces and suffer reproducibility issues, so we mostly used broad adsorbents targeting a range of mid- and lower-volatility gases (Tenax: C6 to C30, Carbotrap 202: C5 to C20). We have revised the text to clarify this point, Section 2.2, lines 136-139:

"Gas sampling cartridges consist of commercially available stainless-steel tubes packed with one or more adsorbing polymer (Tenax® TA, Tenax® GR, Carbotrap® 202, obtained from Sigma-Aldrich). *These sorbents target broad ranges of mid- and lower-volatility gases so are used to validate the samplers across a wide range of gases, including the more moderate-volatility gases that are more likely to suffer sampling losses on surfaces.*"

For other applications, the user of the sampler could choose any adsorbent optimized for their application. We have revised the text as follows to highlight this issue (Introduction, Lines 79-84):

"For sorbent sampling, different sorbents may be used to target various VOCs of interest and large volumes of air can be sampled (Ciccioli et al., 1992; Yokouchi et al., 1990). *Potential users of sorbent tubes should consult the literature and manufacturers to determine the most appropriate sorbent for their needs and target analytes, taking note of the compound types and size ranges adsorbed as well as potential artefacts (Borusiewicz and Zięba-Palus, 2007; Cao and Nicholas Hewitt, 1994; Hwan Lee et al., 2006; Klenø et al., 2002; Rothweiler et al., 1991)*. Advantages of sorbent tubes and TD over whole air samplers are…"

2e) Section 2.4.2: In my opinion, the authors described the complex sampling scheme well, but a simple strip drawing that shows the movement of samplers would make it easier for readers to understand.

We thank the reviewer for the suggestion; a similar comment was made by Referee #1 and we agree the text is confusing. Figure 5 was originally a two-part figure that also included a scheme of sampler movement. We removed it to focus attention on the results grid.

(Section 2.4.2, lines 212-216)

"To examine the utility of distributed sampling to locate emissions sources, the target area was split into a grid of *n* x *n* cells. *Fig. 5 shows a generalized sampling grid with n x n cells and the cardinal directions forming the two sampling angles of 0° and 90°. A three-angle grid would have sampling angles of 0°, 60°, and 120°; a four-angle grid would have angles of 0°, 45°, 90°, 135°, and so on. See Section S4 for further discussion of the number of angles and optimization of the transect mapping approach.*"

[Figure]

**Figure 5: A generalized sampling schematic showing an n x n grid with sampler movement marked by the arrows and oriented in the cardinal directions.**

2f) Section 3.2.: Have the authors tested the ozone capacity (or longevity) of a self-made ozone scrubber? A typical holdup volume of a 25mm syringe filter is about 0.1 mL so it does not hold as much an ozone scrubbing reagent as some of commercially available ozone scrubbers for DNPH carbonyl sampling (e.g., 505285 Supelco LpDNPH Ozone Scrubber). A small discussion about the ozone capacity will be useful here.

We have not tested the longevity of the scrubbers but note that this approach to scrubbing has been previously used for long-term deployment. Our research group has previously this approach to ozone scrubbing for multi-week deployment at 1.5 lpm and demonstrated its continued efficacy, though these data were for a larger diameter filter with several filters stacked. The dimensions used in this work are closer to those used by Ernle et al., 2023 ( https://doi.org/10.5194/amt-16-1179-2023 ) that rigorously investigates the efficacy and lifetimes of sodium thiosulphate scrubbers. Using 37mm glass filters soaked in 10% w/w sodium thiosulfate they determined, at 200 ccm flow rate and 50 ppbv ozone, the filters' lifetimes were 350 hours (see their Table 4) and that lifetime scales linearly with cumulative ozone exposure (i.e., flow rate times time). We can scale these estimates the conditions used here, if we assume that lifetime scales linearly with surface area (our filters have half the surface area), as this would be expected to also scale with the amount of sodium thiosulfate available for reaction.

Under the conditions used here, 25 mm filters saturated with similar sodium thiosulphate concentrations sampled at a flow of 250 ccm, we would expect lifetimes of days (~160 hours) at ozone levels of 40 ppbv, and weeks or months under indoor conditions.

We have not tested the Supelco scrubbers you mention, though these would appear to represent a reasonable approach. Ho et al., 2013 ( https://aaqr.org/articles/aaqr-12-11-tn-0313.pdf ) give a manufacturer's (Waters Corporation, Sep-Pak scrubber) estimate of scrubber lifetime as 48 hrs at 200 ppbv ozone for a flow of 178 ccm, corresponding to a lifetime of 142 hours under our conditions, similar to the approach used here.

(Section 3.2, lines 269-274):

"*This ozone removal approach has the additional benefit of removal particle-phase compounds that may not be removed by the analytical system and could decrease the lifetime of the sampling cartridge. However, it has also been shown to have the downside of potentially removing some oxygenated gases, particularly those with lower volatility (Ngo et al., 2020; Pollmann et al., 2005). Estimates of thiosulphate scrubber lifetimes have been made and effects of humidity also have been studied; these studies observed lifetimes of 14 days at moderate flow rates (200 ccm) and ozone levels (50 ppbv),with longer lifetimes and more efficient scrubbing occurring at higher humidity (80% RH) (Ernle et al., 2023). "*

2g) Tables S2 and S3: Are these numbers the integrated area out of GC/MS? If so, this should be mentioned in the caption.

Yes, these are integrated areas from the GC-MS chromatograms. We should have made it explicit in the captions. Both revised text and captions (in their entirety) are given below:

(Line 18, Supplementary)

"Table S1 shows that 20 of the 31 compounds *had integrated peak areas that* were statistically different between the two sets."

(Table S1, Caption)

"Table S1. The mean and % relative standard deviation (%RSD) *of the areas integrated from the GC-MS* are given for the two sampling sets. Also included are the % increases in the Bench Set (Set A, n=5) versus the Window Set (Set B, n=7). Statistically different analytes have a p-value < 0.05 and are marked with an asterisk(*)."

(Lines 53-54, Supplementary)

"The indoor and outdoor data for the analytes are given in Table S3. Mean *integrated peak area* values are given with their % RSD, and the ratio of unscrubbed/scrubbed is given with an absolute uncertainty."

(Table S3, Caption)

"Table S3. Data for the ozone scrubbing experiments, giving mean values *of the integrated GC-MS areas* for the scrubbed (n=5) and unscrubbed (n=5) sampling sets in both indoor and outdoor settings. A p-value < 0.05 means there is a significant difference between scrubbed and unscrubbed samples. Such values are denoted with an asterisk (*). "